# Waste Material via Geopolymerization for Heavy-Duty Application: A Review

**DOI:** 10.3390/ma15093205

**Published:** 2022-04-29

**Authors:** Marwan Kheimi, Ikmal Hakem Aziz, Mohd Mustafa Al Bakri Abdullah, Mohammad Almadani, Rafiza Abd Razak

**Affiliations:** 1Department of Civil Engineering, Faculty of Engineering—Rabigh Branch, King Abdulaziz University, Jeddah 21589, Saudi Arabia; malmadani@kau.edu.sa; 2Geopolymer & Green Technology, Center of Excellence (CEGeoGTech), Universiti Malaysia Perlis (UniMAP), Arau 02600, Malaysia; rafizarazak@unimap.edu.my; 3Faculty of Chemical Engineering Technology, Universiti Malaysia Perlis (UniMAP), Arau 02600, Malaysia; 4Faculty of Civil Engineering Technology, Universiti Malaysia Perlis (UniMAP), Arau 02600, Malaysia

**Keywords:** geopolymer, waste material, heavy duty application

## Abstract

Due to the extraordinary properties for heavy-duty applications, there has been a great deal of interest in the utilization of waste material via geopolymerization technology. There are various advantages offered by this geopolymer-based material, such as excellent stability, exceptional impermeability, self-refluxing ability, resistant thermal energy from explosive detonation, and excellent mechanical performance. An overview of the work with the details of key factors affecting the heavy-duty performance of geopolymer-based material such as type of binder, alkali agent dosage, mixing design, and curing condition are reviewed in this paper. Interestingly, the review exhibited that different types of waste material containing a large number of chemical elements had an impact on mechanical performance in military, civil engineering, and road application. Finally, this work suggests some future research directions for the the remarkable of waste material through geopolymerization to be employed in heavy-duty application.

## 1. Introduction

In 2000, the world population was over 6 billion people, and it is predicted to grow by 50% in the next half-century, reaching 9 billion in 2050 [1]. Countless products and goods will be delivered via distribution infrastructure to fulfil the requirements and demands of individuals seeking pleasant and convenient lifestyles. As the global economy grows, people began to purchase more items and goods, resulting in an increase in the number of products created and consumed. Solid trash is generated throughout these processes, which is then collected by municipalities and the private waste management industry for recycling or disposal purposes. As society becomes more prosperous, more garbage is produced. Currently, Asia generates roughly one-fourth of the world’s solid waste, although this is predicted to increase to one-third by 2050 [2].

To minimize resources depletion, the Seventh Millennium Development Goal (MDG), which focuses on environmental sustainability through capacity building and sound environmental decision making, calls for the integration of sustainable development strategies into country policies. As a result, one of its key proposals is to Reduce, Reuse, and Recycle, or “3R” [3]. The Society of Solid Waste Management Experts in Asia and the Pacific Island (SWAPI), a network of solid waste management experts, was founded in 2005 with the goal of promoting the 3R’s for solid waste, namely, waste reduction, reuse, and recycling, as well as improving waste management to achieve a 3R Society to conserve natural resources and preserve our living environment.

Economic activity, resource consumption, and economic growth are all intricately related to waste volumes. Economic expansion in Southeast Asian countries is driving annual urban growth rates of 6–8%, a pattern that is likely to last several decades. The trend in waste generation is expected to accelerate as economic developments rise. Table 1 shows the trends in waste generation. Economic trends, demographic projections, and municipal solid waste (MSW) per capita generation rates are used to estimate them. Table 1 demonstrates that in middle-income countries like Malaysia, Thailand, Indonesia and Philippines, waste generation will grow by about 0.3 kg/capita. The growth is mostly due to the prevalence of paper, plastic, bulk waste, and other multi-material packaging in middle-income countries’ waste streams. The waste generation rate in Singapore, a high-income country, is expected to remain relatively steady until falling drastically below its current level.

The waste generation rate in the other nations—Vietnam, Cambodia, Laos, Brunei, and Myanmar—will rise by four to six times the current amount. The density of organic matter and ash residues in waste streams is larger in low-income nations. Additionally, the growing proportion of plastic and paper garbage in the waste stream will contribute to the rising waste volume. In general, the total amount of waste in ASEAN is expected to increase by around 1 million tonnes per day until 2025, compared to existing waste volumes, due to the projected expanding path of economic development [4].

Out of about 300 MtCO_2_e that comes from emerging countries in South and East Asia, the Intergovernmental Panel on Climate Change (IPCC) estimates that landfill methane will reach 1103 MtCO_2_e and 1218 MtCO_2_e in 2020, and 2030, respectively [6]. As a result, proper mitigations must be put in place to prevent future greenhouse gas emission (GHG) from entering the atmosphere. In line with the effort, 3R actions that encourage sustainable waste management often helps to reduce GHG emissions that contribute to the global warming issue. Therefore, as a result of the waste reduction approach, less waste materials were dumped into landfills, reducing the waste material’s degradation potential and, consequently, lowering GHG emissions. This is especially important when dealing with the municipal solid waste (MSW)’s organic component. Diverting organic waste from landfills can minimize the conversion of organic compounds into harmful methane gas, which has 21 times the global warming potential of carbon dioxide.

However, the disposal of these wastes in these landfills, based on present regulations, does not provide efficient and effective management of these solid wastes, which continue to pose a serious environmental danger. Furthermore, a significant amount of waste is still generated each year, with the little land area available to dispose of it. Hence, finding creative ways to value and transform these solid wastes for varied applications would contribute to the implementation of a circular economy and the attainment of a sustainable environment.

### Type of Waste

External factors, such as geographic location, population standard of living, energy sources, and weather, influence waste composition. Quantifying and classifying the various forms of waste created are the most basic stage in waste source management. It is critical to have a system in place for collecting, sorting, and analysing basic waste information, such as the source of wastes, the quantities of waste generated, their composition and characteristic, seasonal variations, and future generation trends. Since municipal, industrial, agricultural, hazardous, and toxic waste, as well was wastewater, require different treatment methods, this is the best way to identify the method to treat waste.

It is possible to find the exact innovation that meets our needs, but we must acknowledge that it will have a significant impact on society and the environment in the future. Researchers are on the lookout for better technology that ensures sustainable development while also protecting our community. As the human population continues to grow, it appears that human needs are increasing as well, resulting in increased demand for food and other necessities. This rising demand also results in waste problem. Agricultural garbage, industrial waste, and domestic waste are polluting the society today, spreading diseases, and destroying nature’s beauty. If this waste is not properly disposed over time, we may not be able to provide a clean and hygienic environment for future generations. It is now our responsibility to appropriately dispose of the waste materials. Garbage can be combined with other materials to be used for various purposes in order to add value to it. Waste materials as reinforcement in composites appear to be a superior option, as it also improves polymer characteristics. Table 2 highlights the various forms of solid waste that are found in our environment and therefore can be effectively utilized.

Municipal solid waste is generated by households, commercial activities, and other sources with activities that are similar to those of households and commercial enterprise, such as waste from offices, hotels, supermarkets, shops, school, and institutions, as well as municipal services like street cleaning and recreational area maintenance. Food waste, paper plastic, rags, metal, and glass are the most common categories of MSW, along with some hazardous household wastes such as light bulbs, batteries, waste pharmaceuticals, and automotive parts.

Industrial waste is a type of trash produced by manufacturing processes such as factories, mill, and mines. It has been yielded since the beginning of the industrial revolution. Most industrial wastes, such as waste fibre from agriculture and logging, are neither harmful nor toxic. From a wide range of industrial processes, the manufacturing industry generates a variety of waste streams. Basic metals, tobacco products, wood and wood products, and paper and paper products are among the most waste-generating industrial sectors in Southeast Asia, particularly in Singapore and Malaysia. In 2000, the Southeast Asian nations contributed to an estimated 19 million tonnes of industrial waste [5]. Meanwhile, in 2010, the Southern American nations passed legislation on “National policy on Solid Wastes” This policy aims to put an end to the disposal of solid waste at open-air dumps, which are placed where waste is simply dumped on the ground [21].

Plastic is a common packaging material, ranging from the well-known disposable plastic carrying bag to the plastic milk bottle. Single-use plastics are a source of concern since they waste a valuable resource when they end-up in landfills. The paper “ The New Plastic Economy” [22] intends to inspire businesses and society to move towards a “circular economy” model for plastic by highlighting impediments to global material flows as well as enablers such as digital technologies [23]. Similarly, the increasing popularity of fiber reinforced polymer composites has been aided by the demand for energy and other limited resources. Vehicles (cars, trains, boats, and planes) can be lighter owing to composites, which improve fuel efficiency. Furthermore, the wind turbine requires lightweight turbines blades, and fibre reinforced composites are an obvious solution. Although composites are long-lasting, waste generated during the manufacturing process is a current concern, and as end-of-life approaches, there will be future concern about ‘disposing’ of massive composites structure. The Composites UK report [24] identifies the recycling alternative for composite materials and compares the environmental impact of various recycling techniques.

Owing to enhanced features such as high specific stiffness, high specific strength, high impact resistance, high abrasion resistance, better corrosion resistance, and higher chemical resistance, polymer matrix composites are widely employed in a variety of applications. They also have low thermal resistance and a high coefficient of thermal expansion. Polymer composites are made up of a polymer matrix with inorganic or organic fillers, which can either be natural or synthetic. Typically, fillers improve the required properties of polymers while also lowering the associated cost. At the time being, due to their improved thermal, mechanical, chemical, and barrier qualities, polymer composites are being used as engineered materials with a variety of applications [25]. Polymer matrix composites are in high demand across a wide range of industries, including aerospace, automobiles, sport, medicine, electronic, civil, communications, energy, construction, industry, marine, military, and various household item applications [26].

Waste material made of geopolymer can be employed to meet the increased demand. Various studies on mixed-based geopolymers are now underway. Previous paper addressed a wide range of recycling waste material to produce advance material as a non-essential application. In contrast, this review will focus on geopolymerization technology in the most often used integrated waste material generation towards heavy duty application. The geopolymerization method with various waste materials can be implemented for the greenhouse gases reduction in the environment. Finally, based on the gaps revealed in the previous literature, additional research opportunities have been proposed.

## 2. Geopolymerization

Geopolymerized composites are currently being studied as a possible replacement for traditional Portland cement-based construction materials. Initially, geopolymer research was limited to natural raw materials such as kaolin, metakaolin, silica fumes, and calcined clays; however, in recent years, the scope of research has expanded to include industrial waste products such as fly ash [27,28], clay-based slag [29,30], palm oil fuel [31] ash, and so on (shown Figure 1) to make them more economically and environmentally sustainable. The fact that practically precursor materials (both natural and industrial waste by-products) emit far less CO_2_ than cement ensures environment sustainability [32]. Considering the high generation (compared to utilization) of industrial waste by-products, disposal concerns, and their harmful/hazardous nature, immobilization/use as a precursor is even environmentally viable. According to current estimates, using geopolymer as a cement substitute in construction products can reduce overall CO_2_ emissions by anywhere from 9% to 64% [32,33]. In fact, these precursor materials are massively generated by industries throughout the world, including fly ash amounting up to 780 million tonnes per year [34,35] (75% to 80% of global annual ash production [36]), palm oil fuel ash, 11 million tonnes per year [37], rice husk ash, 20 million tonnes per year [38,39], red mud alumina, 120 million tonnes per year [40], and the tremendous occurrence of clay kaolin deposits in the earth [41,42]. Furthermore, after accounting for the cost of alkaline activators, the price of geopolymer concrete might be as low as 10–30% lower than conventional cement-based concrete due to reduced price in industrial waste by-products and processing of natural precursor compared to cement.

Geopolymer precursor material must be alumina (Al_2_O_3_) and silica (SiO_2_) component, preferably in reactive amorphous form, in both natural and by-product forms. For the geopolymerization of these aluminosilicate precursors, alkaline activating solutions such as potassium or sodium hydroxide (KOH, NaOH), and potassium or sodium silicate (K_2_SiO_3_, Na_2_SiO_3_) are required. The primary phase begins with the dissolution. In an alkaline media, the species interact ionically, followed by the breakage of the covalent bond between silicon, aluminium, and oxygen atoms. Alkali cations such as Na^+^, Ca^2+^, K^+^, Li^+^, and other charges balance negatively charged ion linked with tetrahedral Al (III). Following that, precursor ions are transported, oriented, and condensed into monomers. Coagulation and gelation are the next steps in the process. Finally, polycondensation of monomers forms rigid 3D networks of silica aluminates [43]. Figure 2 illustrates a conceptual diagram of the several steps of geopolymerization. However, several researchers focus on the variables that could influence the mechanical properties of geopolymer concrete in either a good or negative way. The main disadvantages of geopolymer concrete, as well as the key limitation on geopolymer concrete applications, were found to be the high workability loss rate, short setting time, and the need for heat curing. The type of alkali activator, alkali dosage, fineness of material, and the molar oxide ratios are the most apparent factors that determine the geopolymer properties.

Geopolymer concrete (GPC) is a revolutionary and environmentally friendly concrete that hardens by reacting aluminosilicate waste materials with alkaline activating solutions instead of using cement [44]. GPC allows for a reduction in the requirement for cement production while also providing more outlets for waste materials and industrial by-products. When compared to cement-based concrete, it is projected that using GPC might save up to 43% on energy and lower greenhouse gas emission by 9–80% [45]. This wide range is owing to the complexities of calculating emissions, which depend on a number of factors such as local conditions, transportation, and the mix design itself [19]. In addition, compared to regular concrete, GPC has better durability features, such as chloride resistance [46], high temperature resistance [47,48], freeze-thaw cycles [49], and carbonation resistance [50]. It has been demonstrated that GPC has appropriate compressive and tensile strength [51].

## 3. Waste-By Products Based Geopolymer

Fly ash is a by-product of the manufacturing process of coal combustion that is split into two classes: class F and class C. The combustion of bituminous coal creates a king of fly ash known as class F fly ash, which has a very low CaO level (FFA). Class C fly ash with high calcium content is also produced using lignite and sub-bituminous coal as new power sources [52]. FFA has a similar composition to natural volcanic ash [53]. Fly ash is a readily available by-product with a microscopic shape of small spherical particles that is commonly utilized as a raw material for manufacturing geopolymer [43,54]. The high free-CaO level of CFA limits its use in the OPC system, and its use in geopolymer preparation has been beyond imagination [55]. The chemical composition of various raw materials is shown in Table 3, FFA and CFA have Si/Al ratios of 1.86–3.09 and 1.82–2.52, respectively. Fly ash has been used in cement and concrete since the early 20th century, and it is often used as a major component. It is better for the environment to use FA instead of cement because it minimizes greenhouse gas emission and construction budgets. FFA has a reasonable price, is readily available, has a nice spherical structure, and the aluminate and amorphous silicate have a high activity. In alkali activator solution, high-strength geopolymers can be easily generated [56].

Due to the existence of amorphous phases, high hardness, and pozzolanic activity, ground granulated blast furnace slag (GGBS) is mostly utilized as a partial alternative of OPC after grinding, depending on the cooling condition [57]. Table 3 shows that GGBS is extremely reactive in geopolymers synthesis, and a satisfactory reaction rate can be achieved at temperatures as low as room temperature. When slag is utilized as a cement alternative, it produces less heat during hydration, which reduces the risk of cracking [58]. GGBS can be utilized in a variety of situations such as to enhance concrete porosity, long-term strength, and resistance to sulphate and alkali silicate reactivity, as well as hydration heat, permeability, and lower water demand [59,60].

The Bayer process, which is employed in industrial aluminium refining, produces red mud (RM) as a by-product. The Bayer method dissolves the soluble component of bauxite with sodium hydroxide at high temperatures and pressures. A small quantity of sodium hydroxide employed in this method will invariably remain in the RM, leading to higher pH value [61]. By eliminating the need for mud drying, using RM in the form of mud saves time and energy. It also reduces the total amount of alkali activator by utilizing high alkalinity red mud, thus lowering the cost of geopolymer manufacturing [62]. The appropriate replacement value of RM for FA-based geopolymers varies depending on NaOH concentration and curing conditions [63]. Furthermore, the geopolymer mixed with red mud has increased strength and durability, according to the research by Liu et al. [64].

**Table 3 materials-15-03205-t003:** Chemical composition of various waste by-product geopolymers.

Type of Slag	Chemical Composition (wt %)							
	SiO_2_	Al_2_O_3_	CaO	MgO	Fe_2_O_3_	K_2_O	Na_2_O	SO_3_
Fly Ash [65]	55.38	28.14	3.45	1.85	3.31	1.39	2.30	0.32
Fly ash [66]	56.00	18.10	7.24	0.93	5.31	1.36	1.21	1.65
Fly ash [67]	65.90	24.00	1.59	0.42	2.87	1.44	0.49	N/A
Fly ash [68]	47.90	25.70	4.11	1.36	14.70	0.67	0.81	0.19
High Calcium Fly Ash [69]	37.30	14.90	17.10	3.72	16.50	1.66	1.74	2.56
High Calcium Fly Ash [70]	34.00	13.50	16.50	3.10	5.00	5.50	1.50	2.80
High Calcium Fly Ash [71]	36.20	19.90	14.20	1.90	11.90	2.40	N/A	3.60
Ground Granulated Blast Furnace Slag [72]	35.34	20.69	31.32	8.11	0.18	0.29	1.36	1.79
Ground Granulated Blast Furnace Slag [73]	18.90	6.43	66.90	1.41	0.74	0.67	N/A	1.97
Ground Granulated Blast Furnace Slag [74]	28.20	9.73	52.69	2.90	0.98	1.22	N/A	1.46
Ground Granulated Blast Furnace Slag [75]	36.50	9.95	43.38	6.74	0.38	0.35	N/A	N/A
Red Mud [76]	14.40	22.20	2.00	0.17	40.20	0.11	12.70	0.28
Red Mud [77]	16.51	28.05	2.22	0.70	30.32	0.26	8.70	N/A
Red Mud [78]	27.54	30.59	25.48	0.49	4.60	N/A	N/A	1.42
Rice Husk Ash [79]	92.33	0.18	0.63	0.82	0.17	0.15	0.07	N/A
Rice Husk Ash [80]	93.10	0.30	1.50	0.49	0.20	2.30	0.06	N/A
Silica Fume [79]	87.60	0.38	0.57	3.67	0.66	2.36	1.26	N/A
Silica Fume [81]	90.00	1.20	1.00	0.60	2.00	N/A	N/A	0.50
Volcanic Ash [82]	43.32	14.84	8.80	7.70	14.19	1.52	3.04	0.01
High Magnesium Nickel Slag [74]	43.22	4.35	3.45	26.15	10.34	0.18	0.23	0.28

Another waste by-product is rice husk ash (RHA) that is produced from rice husk combustion. RHA, a silica rich agricultural waste, is regarded as a clean alternative for improving the characteristic of geopolymers [83]. The use of RHA in geopolymer concrete can reduce nano-SiO_2_ consumption and pollution issues caused by RHA disposal in landfills, particularly in rice-producing countries [84]. RHA has been widely used in self-compacting geopolymer concrete due to its greater reactivity inspired by high silicon concentration and ultra-high specific surface area [85]. Sugarcane bagasse ash is an industrial by-product that has been used as a source of alumina and silicates in volcanic as a product by a number of researchers [86].

The principal by-product of municipal solid waste incineration is bottom ash. Heavy metals are abundant in the bottom ash, which has a small particle size [87]. In recent years, bottom has been increasingly recycled as building binders and concrete [84,88]. Moreover, bottom ash from the burning of municipal sewage sludge is employed in concrete at a concentration of 10–15%, resulting in greater strength than concrete without bottom ash [83]. High silica and alumina concentrations can be found in fly ash, blast furnace slag, red mud, and materials such as rice husk as main biomass ash, making them appropriate as supplementary materials for gelling. Steel slag, volcanic ash, silica fume, waste glass, coal gangue, high-magnesium nickel slag, and other minerals are also often employed. Numerous industrial catalyst residues include enough silicon and aluminium, as well as an amorphous structure that can be used to form synthetic geopolymers, and its compressive strength has been measured to be between 40 to 85 MPa [89]. It is obvious that the raw materials used to discover geopolymer are high in silica and aluminium, and calcium oxide content cannot be neglected.

In conclusion, the raw materials might be an aluminosilicates natural mineral including silicon, aluminium, oxygen, and other possible elements. The right raw materials should have amorphous properties and a high ability to release aluminium easily.

## 4. Heavy-Duty Applications of Geopolymers

Geopolymer applications can be divided into two groups based on their function: those having varied physical and chemical properties as well as those with physical and mechanical properties. Buildings such as fire prevention buildings, insulation walls, and nuclear power plant can make use of these functional applications for fire prevention, isolation, heat preservation, and adsorption of hazardous ions. Table 4 shows the utilization of waste material-based geopolymer in heavy-duty applications.

The stability and safety of a structure will be compromised if it is exposed to rains, ocean, or saline soil over an extended period of time. However, the chemical resistance of geopolymer concrete, particularly sulphate resistance, makes it more suitable for marine building. Geopolymer concrete is comprised of more amorphous phases, smaller porosity, and more mesopore than OPC concrete, and the dense microstructure of geopolymer concrete makes seawater permeation harder [96,97]. When compared to OPC concrete, geopolymer concrete has greater chloride ion erosion resistance and a longer corrosion cracking time, making it an excellent prospect for use as an anti-corrosive coating in the maritime environment [98]. According to Chindaprasirt and Chalee [99], the penetration and corrosion of chloride ion reduced as the molarity of sodium hydroxide increased after the fly ash-based geopolymer was exposed to the tidal zone of the ocean environment for three years after being air-dried in the laboratory for 28 days. Nevertheless, after six years in a salt lake environment, fly ash-based geopolymer concrete is more easily carbonized than OPC concrete, and chloride and sulphate are more easily diffused [100]. However, according to Alzeebaree et al. [101], both carbon fibre and basalt fiber reinforced geopolymer fabric can be employed as the modification material to resist chloride ion erosion. Additionally, fibre reinforced geopolymer concrete allows it to be employed as a structural member instead of ordinary concrete. The permeability of chlorine ion can be reduced by adding OPC to fly ash, whereas the permeability of chlorine ion can be strengthened by introducing metakaolin and nano-SiO_2_ [102].

### 4.1. Geopolymer in Military Application

Geopolymer was used in the heavy duty rigid pavements (turning node, aprons, and taxiways) at a commercial airport Brisbane, Australia [103], as well as the Global Change Institute (GCI) building at the University of Queensland [104]. Pre-stressed geopolymer concrete sleepers have previously been produced by one of Australia’s leading concrete sleeper providers and have been successfully used on mainline railway tracks [105]. Considering geopolymer concrete as having better acid resistance and less alkali-silica reaction than typical OPC concrete, it has been recommended for use in the construction of railway sleepers, which are exposed to chemicals and prolonged environmental circumstance [106]. Furthermore, the US Army Corps of Engineer’s Waterways Experiment Station (WES) stated that fly ash-based alkali activated aluminosilicate binder can be potentially used to repair deteriorate Army airbase concrete and other special construction demands.

According to previous reports, the US Air Force and Navy, the Royal Air Force (RAF), and the Royal Australian Air Force (RAAF) have all had concrete durability difficulties with their F/A-18 and B-1 parking aprons [107,108,109]. Military airfield concrete, particularly aprons, has been exposed to severe thermal shocks from jet exhaust and has been discovered saturated with chemical like hydrocarbon fluids (HF); aircraft engine oil, hydraulic fluids, and jet fuel [107]. During engine start-up, the surface temperature of the apron concrete underneath F/A-18 auxiliary power units (APUs) can reach 175 °C in 10–12 min [110]. Figure 3a shows an ancient deep scaling where particles were scraped away from the concrete’s wearing surface while Figure 3b depicts an APU in the bottom of an F/A-18 [107]. This type of damage is a source of foreign object debris (FOD), and it is more common in the area where the Auxiliary power unit (APU) exhaust impinges on the concrete. A substantial amount of split engine oil, hydraulic fluid, and vented jet fuel from the aeroplane is also commonly observed in similar areas of pavement where the APU exhaust impinges concrete. It is also worth noting that the jets tend to park in roughly the same spot each time, causing localised damage to some aprons. In the construction of rigid pavements at military airbases, several researchers have proposed substituting OPC with other heat and chemical resistant cementitious materials [111]. The feasibility of geopolymer binder for replacing OPC at military airbases should be examined because it is more resistant to both heat and chemicals and is more durable than OPC. Hence, it can be concluded that geopolymer can be a promising alternative to OPC for repairing apron concrete at military airbases.

### 4.2. Geopolymer in Civil-Engineering Application

Structures that are still in the design phase are likely to be subjected to blast and impact loading threats. Due to its high-ultra strength, high ductility, and outstanding toughness, Portland cement-based ultra-high performance concrete (PC-UHPC) has been developed in recent decades to meet the increasing safety requirements of structures to overcome such destructive intensive loadings. Although PC-UHPC has emerged as one of the most promising construction materials for civil and military structures, the extensive use of Portland cement has a negative impact on the environment due to the carbon dioxide emission produced during cement manufacture. Meanwhile, the environment is facing a large increase in the formation of industrial wastes and the consumption of raw material in cement manufacture. As a result, it is required to develop a geopolymer as an alternative binder system that is less expensive and energy-intensive while also being greener in order to reduce or fully replace ordinary Portland cement. However, it has been noted that the higher the geopolymer’s strength, the greater the fire resistance. Low-strength and low-density geopolymers are difficult to dehydrate and react to volume changes better in the temperature range of 100 °C to 1000 °C; even after heat exposure, their intensity increases [112]. Figure 4 shows the effect of alkali cation selection on the fly ash-based geopolymer’s high temperature exposure strength and durability. Depending on the type of alkali cations utilized, densification of particular substrates and healing of microcracks are useful to increasing strength in different temperature ranges [113]. These findings imply that geopolymers can be tailored to attain stable (and even improved) strength after exposure to a high heat environment. The building’s damage caused by fire cannot be ignored. The Windsor Tate Fire in Shanghai, as well as the 9/11 terrorist attack, resulted in massive human and material losses. As a result, refractory materials for building are crucial. Today, continuing to improve the sustainability and ecology of fire-resistant and high-temperature materials is a primary concern. Therefore, the goal of geopolymerization is to turn industrial solid waste into a chemically durable cement binder that is both thermally stable and non-combustible.

### 4.3. Geopolymer in Road Application

The development of geopolymers in the past and present has centred on the production and use of such materials to replace cement in structural construction. There has not been a lot of research applied in road construction. Several geopolymer research reports for road applications were even at the proof-of-concept level. Tenn et al. [114] investigated the interaction between sodium and potassium-based geopolymer binders and granite or diorite pavement aggregates in order to promote the usage of geopolymer in place of asphalt cement (or bitumen). Camacho-Tauta et al. [115] demonstrated an attempt to improve road fatigue damage resistance by employing a fly ash-based geopolymer as a road base layer. To assess the study material’s long-term performance, a full-scale accelerated pavement test was assigned. Compared to a non-treated road base layer, the study found that a geopolymer-treated road base layer could give a reduction in deformability. Waste material-based geopolymer has been implemented as a replacement binder to enhance the properties for road and pavement material as tabulated in Table 5.

The Netherlands was one of the first countries in Europe to use fly ash and blast furnace slag as a binder for acid-resistant pipe manufacture. Activated alkaline materials used in civil construction subsequently established enterprises in other United Kingdom countries, and eventually extended throughout the continent [120,121]. Conversely, the company with the most building applications is based in Australia. In 2007, the Melbourne-based company, E-cert, developed its own concrete. This company employs a blend of fly ash and ground blast furnace slag that has been alkali activated according to a proprietary dosage and composition. Bridges, highways, and big structures are several of the applications [122].

Figure 5a depicts a section of the Westgate Freeway in Port Melbourne’s highway pavement. Since it was a different material, the project had to comply to numerous needs of the local road authority as well as more specialized technical standards in order to gain government clearance. The highway’s construction and use were agreed on by a group of multinational construction corporations. Meanwhile, Figure 5b illustrates VicRoad’s installation of 55 MPa E-Crete prefabricated panels. Due to the strict inspection in regard to structural concrete, this strength was required [122].

## 5. Conclusions and Suggestions for Future Works

Regardless of the differences in waste material used in geopolymerization, the selection of raw materials depends on the desired application. Waste material-based geopolymer rich in silicon, aluminium, and calcium are easy and highly accessible. This facilitates their disposal waste management rather than building a landfill full of them. This is the main reason behind the low cost of manufacturing geopolymeric composites-based waste material in civil and military application. The utilization of waste materials through geopolymerization should be carried out in future works to mitigate the disposal and environmental issues. Numerous studies have been conducted involving the use of waste-based geopolymers with promising properties for heavy-duty applications such as military, civil-engineering, and road applications.

Based on the identified gaps in this work, future recommendations on waste-material based geopolymer in heavy duty application are listed below:Durability works using waste material in advanced application in the civil construction or aerospace fields;Establishing standards in order to conduct more advanced tests and research on these waste materials and, as a a result, expand the application as heavy-duty material in civil construction or the aerospace industry;Besides construction and airbase application, the geopolymer material can also be implemented as a defence material that consists of lightweight and higher mechanical properties such as bulletproof, Kevlar helmet, and body armour;The study on the landfill and waste management cost is crucial in considering the impact of the 3R implementation;An alternative activator to hydroxides and silicates that leads to lower environmental impact and can cut the cost of geopolymer production;For better understanding and experimental application, standardize dosage and quantify ingredients utilized in the manufacturing of activated alkali component.

## Figures and Tables

**Figure 1 materials-15-03205-f001:**
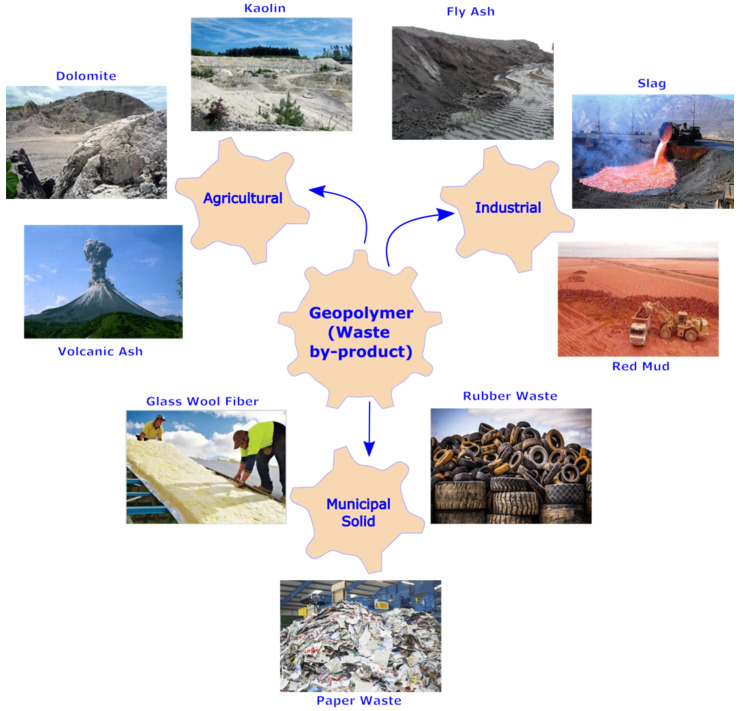
Common type of precursor materials used in geopolymer material.

**Figure 2 materials-15-03205-f002:**
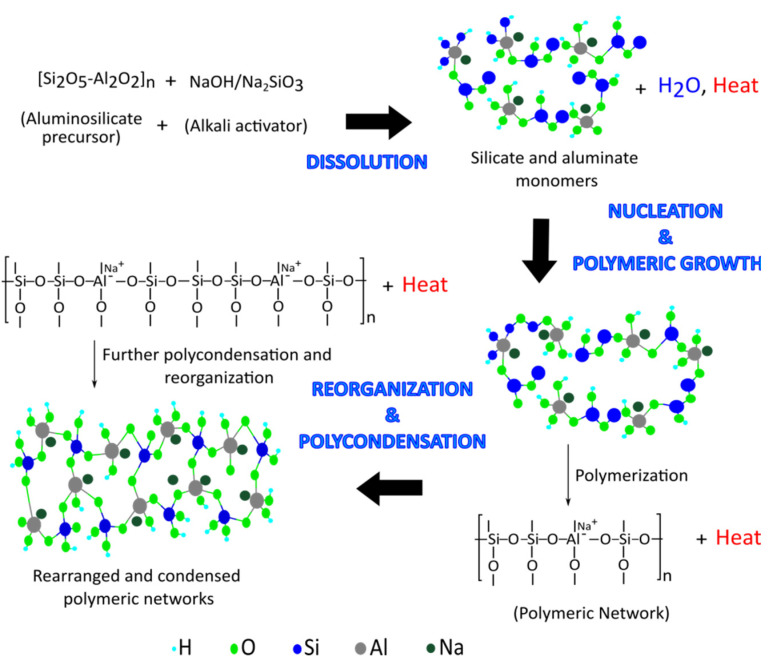
Conceptual process of geopolymerization.

**Figure 3 materials-15-03205-f003:**
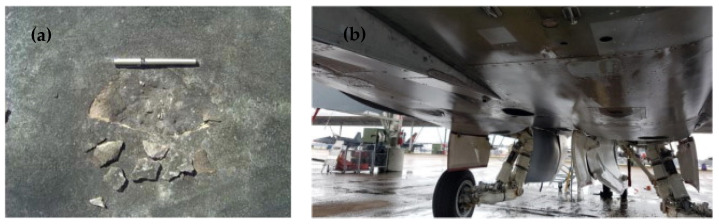
(**a**) Scaling at the top layer of the military airbase concrete and (**b**) Underbelly of an F/A-18 with the APU in the centre. Reprinted/adapted with permission from [107]. 2018. Shill.

**Figure 4 materials-15-03205-f004:**
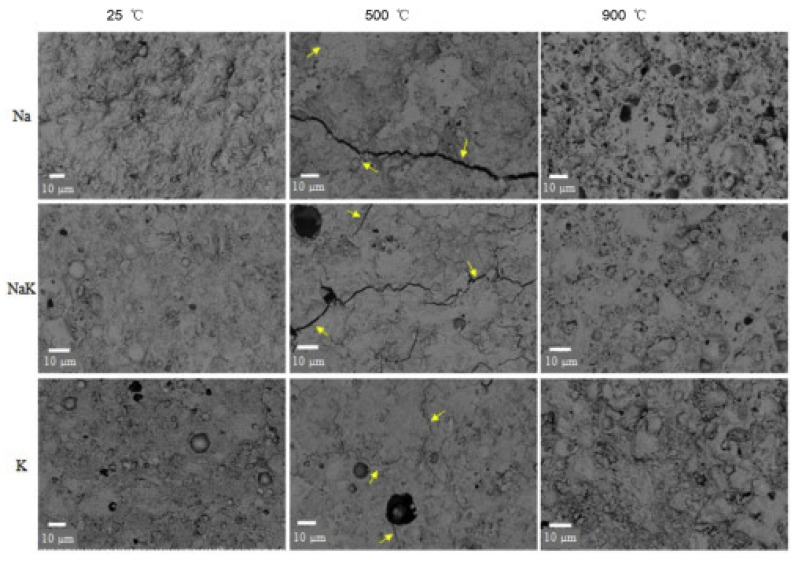
SEM-BSE micrographs depict the formation and healing of micro-cracks in geopolymer at various temperatures (solid arrows showing micro-crack). Reprinted/adapted with permission from [113]. 2018. Lahoti.

**Figure 5 materials-15-03205-f005:**
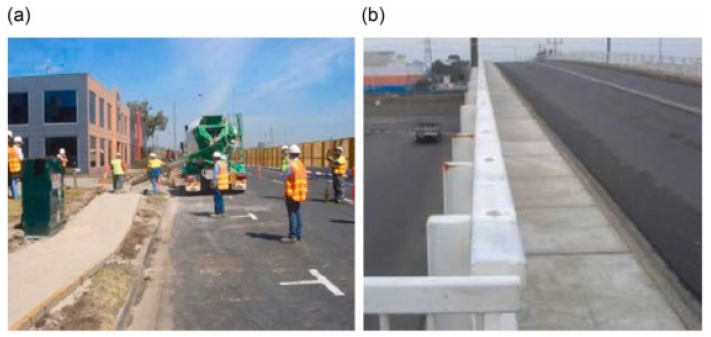
(**a**) Westgate Freeway paving in Port Melbourne and (**b**) E-Create Precast Panels. Reprinted/adapted with permission from [122]. 2012. Van Deventer.

**Table 1 materials-15-03205-t001:** The rate of municipal solid waste generation per capita in urban ASEAN by 2025. Reprinted/adapted with permission from [5]. 2009. Ngoc.

Country	Gross National Product Per Capita (USD)		Waste Generation Rate (kg/cap/day)		Predicted Urban Waste Generation	
	1995	2025	Generation Rates (kg/cap/day)	Total Waste (tons/day)	Municipal Solid Waste (kg/cap/day)	Total (tons/day)
High Income						
Singapore	26,730	36,000	1.1	4840	1.1	4840
Middle Income						
Thailand	2740	6700	0.64	15,715	1.5	3673
Indonesia	980	2400	0.76	96,672	1.0	1272
Philippines	1050	2500	0.52	33,477	0.8	5150
Malaysia	3890	9440	0.81	15,663	1.4	2681
Low Income						
Vietnam	240	950	0.61	19,983	1.0	3276
Brunei	260	750	0.66	149,140	0.95	2169
Cambodia	220	700	0.52	3544	1.1	7497
Myanmar	240	580	0.45	12,118	0.85	2289
Laos	350	850	0.55	1379	0.9	2257

**Table 2 materials-15-03205-t002:** Solid wastes and related possible uses are described in detail.

Type of Waste	Sources of Content	Potential Application	References
Hazardous Waste	Trash from galvanising, tannery waste, and metallurgical waste	Cement brick, tiles, boards	[7,8]
Mining Mineral Waste	Overburden waste tailing from the iron, coal wateriest waste, copper, gold, zinc and aluminium industries	Light-weight aggregate fuel, brick, tiles	[9,10]
Agro Waste	Cotton stalks, husk from packed rice and wheat straw, sawmill waste, jute and banana stalks, nut shells, sisal, and vegetable residue	Insulation boards, particle board, wall panel, roofing sheets, fibrous construction panel, fuel binder, acid resistant cement	[11,12,13]
Industrial Waste	Bauxite red mud, steel slag, construction detritus, coal combustion residues	Bricks, blocks, cement, paint, wood substitutes, tiles, concrete, and ceramic goods	[14,15,16]
Non-hazardous Waste	Gypsum waste, lime sludge limestone waste, marble production waste,	Cement clinker, super sulphate hydraulic binder, gypsum plaster, fibrous gypsum, boards, bricks and blocks	[17,18]
Municipal Solid Waste	Soft drink bottle, jar for food, cosmetics product	Replacement binder material, supplementary material in concrete, soil stabilization	[19,20]

**Table 4 materials-15-03205-t004:** The potential geopolymer material and possible application are described in detail.

Geopolymer Waste Material	Potential Application	Properties	Ref.
Ground granulated blast furnace slag, Fly ash, granite coarse aggregate	Concrete pavement	50 MPa of compressive strength and 4.72 MPa of flexural strength	[90]
Red mud waste (bauxite residue), slag	Heavy metal removal, composite materials, Adsorbent and coagulant	66 to 86 MPa of Compressive strength	[91,92]
Ferrosilicon slag, alumina waste	Thermal insulation brick	10.9 MPa of compressive strength and 0.59 W/m.k of thermal conductivity	[93]
Metakaolin, bottom ash waste	Thermal insulation brick	47.9 MPa of compressive strength, 1.32 W/m.k of thermal conductivity	[94]
Blast furnace slag, rice husk ash	Acid proof cement	57 MPa of compressive strength	[95]

**Table 5 materials-15-03205-t005:** Research work utilising geopolymers in road applications.

No.	Researcher	Materials	Findings
1	Sukprasert et al. [116]	Fly ash, silty clay, ground granulated blast furnace slag	Increase packing density Increased unconfined compressive strength
2	Dave et al. [117]	Ground blast furnace slag, fly ash, silica fume	Appropriate strength as road repair material Well durability through ultrasonic pulse velocity test
3	Wongsa et al. [118]	Crumb rubber, river sand, high calcium fly ash	Average value of thermal conductivity and density Meet the strength requirement for lightweight concrete
4	Mohammed et al. [119]	Fly ash, crumb rubber	Reduction in compressive and flexural strength Higher water absorption

## Data Availability

Not Applicable.

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
