# Peer review of "Waste Material via Geopolymerization for Heavy-Duty Application: A Review"

_materials, 2022, doi:10.3390/ma15093205_

Round 1
Reviewer 1 Report
- The publication submitted for review is a literature review, a much needed compilation for scientists and the military industry and another. The review exhibited that different types of material containing a large number of chemical elements had an impact on mechanical performance, particularly in heavy-duty application. A comprehensive literature study and critical evaluation on crystallographic, chemical bonding analysis, and microstructure analysis, and remarkable future development of waste material via geopolymerization technology heavy-duty applications is conducted in this review.
- The article is an inspiration to businesses and society to move towards a circular economy‛ model for plastic by highlighting impediments to global material flows as well as enablers such as digital technologies.
- However, the publication needs to be put in order. In the version presented for review after chapter 1, we have chapter 3, then 2, and then 3. It needs to be corrected.
- The introduction is very extensive and sometimes differs from the topic.
- Since the authors describe the various applications of geopolymers very broadly, they should include it in the title. Then my comment from point 4 is unfounded.
- I suggest you consider changing the title of the publication.
- Since the research on geoplolymers is very advanced and different, in my opinion, such articles presenting collectively the latest information on this topic are very necessary.
- Apart from the minor mistakes I mentioned earlier, the review - publication is prepared very carefully.
Author Response
- The publication submitted for review is a literature review, a much needed compilation for scientist and the military industry and another. The review exhibited that different type of material containing a large number of chemical elements had an impact on the mechanical performance, particularly in heavy-duty application. A comprehensive literature study and critical evaluation on crystallographic, chemical bonding analysis, and microstructure analysis, and remarkable future development of waste material via geopolymerization technology heavy-duty applications is conducted in this review.
Response: Thank for your valuable comment.
- The article is an inspiration to business and society to move towards a circular economy’s model for plastic by highlighting impediments to global materials flows as well as enablers such as digital technologies.
Response: We gratefully appreciate for your valuable suggestions.
- However, the publication needs to be put in order. In the version presented for review after chapter 1, we have chapter 3, then 2, and then 3. It needs to be corrected.
Response: We are grateful for the reviewer for noticing this. We have made an extensive editing for the chapter order. We have highlighted the changes in our revised manuscript as follows (Line 96, 183, 245, 314, 348, 396, 432, 473, 486)
- The introduction is very extensive and sometimes differs from the topic.
Response: Thank for you kind suggestion. We have made the corresponding revision, including restructure the introduction section (chapter 1).
- Since the authors describe the various applications of geopolymers very broadly, they should include it in the title. Then, my comment from point 4 is unfounded.
Response: We are sorry for the inconvenience brought to the reviewer. We apologize for our inaccurate description. To facilitate your understanding, we have changes the title.
- I suggest you consider changing the tittle of the publication.
Response: Thank you for your constructive suggestion. We apologize for our inaccurate description. To facilitate your understanding, we have changes the title.
- Since the research on geopolymers is very advanced and different, in my opinion, such articles presenting collectively the latest information on this topic are very necessary.
Response: We appreciate the reviewer commenting on this. We added the latest information from previous studies which are related on this current topic. (Table 4 and 5, Section 4.2)
- Apart from the minor mistakes I mentioned earlier, the review- publication is prepared very carefully.
Response: We highly appreciated the reviewer’s positive appraisals of our manuscript.

Reviewer 2 Report
The paper "Waste material via geopolymerization for heavy-duty towards military application: A review" addresses an important topic related to the application of geopolymer materials in specific applications, such as military defense. The topic is innovative and fully adherent to research published in Materials, and may be considered after further corrections:
(1) The abstract must be finished with a sentence that indicates future perspective and/or gaps in the study literature;
(2) The issues addressed in the first sections should be supplemented. The authors mention waste types issues, and many of the results are generic. Some more current research should be considered on this topic, such as: 10.1007/s10668-021-01630-7; 10.1016/j.cscm.2021.e00637.
(3) In section 2, the authors address the process of geopoliemrization, they should complement with disadvantages of the use of geopolymers! the central question is why this material is still little used in view of the countless advantages? This should be very clear to readers.
(4) The flowchart in Figure 1 can be complemented with images;
(5) Think of sections 3 and 4 as a complement to other studies that used waste materials in geopolymers, such as: 10.1016/j.cscm.2021.e00839, and others that you can consult at: https://www.sciencedirect. com/journal/case-studies-in-construction-materials/special-issue/100C1LHWPSX. As a review article it must be very current and with cutting edge research, this will indicate future citations in your research;
(6) It lacks comparative tables of its studies with others in the literature, this is essential in a review work;
(7) The concluding section should be separated from the future perspective of new research, which should be highlighted and show the challenges of its implementation.
Author Response
- The paper “Waste material via geopolymerization for heavy-duty towards military application : A review” addresses an important topic related to the application of geopolymer materials in specific applications, such as military defense. The topic is innovative and fully adherent to research published in Materials, and may be considered after corrections:
Response: Thank for your valuable comment. We highly appreciated the reviewer’s positive appraisals of our manuscript.
- The abstract must be finished with a sentence that indicates future perspective and/or gaps in the study literature
Response: Thank you for your considerable comments. We are apologizing for our lack description. We have included the future perspective in the study literature.
- The issues addressed in the first sections should be supplemented. The authors mention waste types issues, and many of the results are generic. Some current research should be considered on this topic, such as 10.1007/s10668-021-01630-7; 10.1016/j.cscm.2021.e00637
Response: We are sorry for the inconvenience brought to the reviewer. To facilitate your understanding, we added the corresponding new reference in the revised manuscript as suggested. (Section 4.2, Table 5)
- In section 2, the authors address the process of geopolymerization, they should complement with disadvantages of the use of geopolymers. The central question is why this material is still little used in view of the countless advantages? This should be very clear to readers.
Response: Thank you for your considerable comments. We are apologizing for our lack description. We have made the corresponding revision, including addition new sentence as follow (Line 222-228)
- The flowchart in Figure 1 can be complemented with images.
Response: Thank for you constructive suggestion. To address the reviewer’s concern, we replace the Figure 1.
- Think of sections 3 and 4 as a complement to other studies that used waste materials in geopolymers, such as 10.1016/j.cscm.2021.e00839, and others that you can consult at: https://www.sciencedirect.com/journal/case-studies-in-construction-materials/special issue/100C1LHWPSX. As a review article it must be very current and with cutting edge research, this will indicate future citations in your research.
Response: We appreciate the reviewer’s insightful suggestion and agree that the article must be very current and with cutting edge research. Hence, we have include the new reference as follows (Chapter 4, Table 4)
- It lacks comparative tables of its studies with others in the literature, this is essential in a review work;
Response: We appreciate the reviewer’s insightful suggestion and agree that the comparative table is lack in the article. We have added the comparative table for better understanding in the revised manuscript (Table 4 and 5)
- The concluding section should be separated from the future perspective of new research, which should be highlighted and show the challenges of its implementation.
Response: Thank for you constructive suggestion. To address the reviewer’s concern, we refine the conclusion. Also, we highlighted the gaps and future works on the future perspective of new research and challenges of its implementation. (Chapter 6)

Round 2
Reviewer 2 Report
Ok